# Evaluation of the Burdening on the Czech Population by Brominated Flame Retardants

**DOI:** 10.3390/ijerph16214105

**Published:** 2019-10-24

**Authors:** Hana Logerová, Petr Tůma, Michal Stupák, Jana Pulkrábová, Pavel Dlouhý

**Affiliations:** 1Department of Hygiene, Third Faculty of Medicine, Charles University, Ruská 87, 100 00 Prague 10, Czech Republic; hana.logerova@lf3.cuni.cz (H.L.); pavel.dlouhy@lf3.cuni.cz (P.D.); 2Department of Biochemistry, Cell and Molecular Biology, Third Faculty of Medicine, Charles University, Ruská 87, 100 00 Prague 10, Czech Republic; 3University of Chemistry and Technology Prague, Faculty of Food and Biochemical Technology, Department of Food Analysis and Nutrition, Technická 3,16628, Prague, Czech Republic; michal.stupak@vscht.cz (M.S.); jana.pulkrabova@vscht.cz (J.P.)

**Keywords:** polybrominated diphenyl ethers, DDTs, flame retardants, human fat tissue, gas chromatography, mass spectrometry, PCBs

## Abstract

The completed environmental study was concerned with assessing the exposure of the Czech population to polybrominated diphenyl ethers (PBDEs). Simultaneously, the levels of polychlorinated pollutants such as polychlorinated biphenyls (PCBs) and chlorinated diphenyl ethanes (DDTs) were also monitored. The pollutant levels were newly measured in solid fat tissue removed during plastic surgery. A total of 107 samples of fat were taken from 19–76-year-old volunteers. A total of 16 PBDE congeners were determined, of which only six occur in more than 38% of fat tissue samples. The total PBDE level attains an average value of 3.31 ng/g, which is 25% less than was measured in 2009. On the other hand, there was an increase in the levels of two PCB congeners, which was caused by an increase of the total PCB concentration from level of 625.5 ng/g, published in 2009, to the current level of 776 ng/g. The level of DDTs decreased and currently has a value of 467.4 ng/g, which is about 24% lower than in 2009. The contamination of obese middle-aged women in Czechia by more modern types of pollutants, such as PBDEs, is incomparably lower than that by PCBs and DDTs and is also decreasing in time.

## 1. Introduction

Industrial production of flame retardants began in the 1970s in the USA and Japan. Flame retardants were incorporated into polymer plastic matrices to reduce the flammability of these materials and the risk of fires [1]. Their use became widespread in Czechoslovakia at the beginning of the 1990s. According to their chemical structures, these substances are divided into two groups: inorganic retardants, which do not constitute a substantial risk for the environment and human health [2], and organic retardants containing halogen atoms [3] or phosphorus (phosphorus flame retardants were responsible for 20% of total flame retardants consumption in 2006 in Europe) [4], which constitute a substantial burden for the environment and a potential health risk for humans. Organic retardants are now found in all the components of the environment, which they contaminate during the production, use, and disposal of plastic products after the end of their lifetimes. 

At the present time, the commonest group of flame retardants consists of polybrominated diphenyl ethers (PBDEs), which are incorporated into polymeric matrices only through weak physical interactions so that they are relatively easily released at elevated temperatures or when exposed to intense solar radiation [5]. PBDEs represent a group of 209 different chemical structures designated as congeners (Figure 1); PBDEs = C_12_H_(10-x)_Br_x_O, *x* = 1, 2 to 10; the number of individual isomers of Mono, Di, Tri to DecaBDE are 3, 12, 24, 42, 46, 42, 24, 12, 3, and 1, respectively. Industrially manufactured PBDEs always represent a mixture of several congeners and are sold under various commercial names, such as DE-60F, DE-71, DE 83R, and a great many more. For example, the product designated as PentaBDE contains, in fact, 50–62% PentaBDE, 24–38% TetraBDE, 4–8% HexaBDE, and 0–1% TriBDE. The most dangerous are PBDEs with fewer bromine atoms (1–4), which are more effectively accumulated in the environment and their detrimental effects on the synthesis of thyroid hormones, reproduction, fetal development, and liver function have been described, and they also constitute a neurological risk [6,7,8,9,10].

The human population is exposed to PBDEs through ingestion of food like fish and shellfish [11,12], meat products and eggs [13,14], dairy products, and oils [15]. Instead of food, the ingestion of dust from the home environment represents the highest intake of BDE-209 [16,17]. Infants are exposed to PBDEs via breast milk [18]. An additional exposition pathway represents the inhalation of airborne dust particles in the indoor and outdoor environment [19,20]. Important exposure also occurs during long-term breastfeeding when PBDEs are released from the fat tissue into the breast milk and blood, similar to PCBs (Figure 1) [21,22].

All the mentioned types of contamination are connected with the nonpolar nature of PBDEs and their related very strong bioaccumulation potential. PBDEs are accumulated especially in white fat tissue, from which they are gradually released during long-term starvation. PBDEs were also identified in blood plasma, where they are transferred by bonding to plasma proteins, similar to thyroid hormones [23]. For all these reasons, the production and use of a number of PBDEs and other flame retardants or pesticides, such as hexabromobenzene (HBB), PentaBDE, OctaBDE, and Lindane, are regulated by the Stockholm Convention [24]; in addition, the use of technical mixtures of PBDEs, with the exception of DecaBDE, was prohibited in EU countries by an EU Directive of 15 August 2004 [25].

Globally, PBDEs were demonstrably detected in breast milk [9,26], blood [9,27], fat tissue [28,29,30], and fetal livers [31]. Epidemiological studies have described the dependence between exposure to PBDEs and reduced thyroid function [32], spermatogenesis disorders [33], and endocrinal disruptions [34,35]. Monitoring of PBDEs in Czechia was commenced at the beginning of the 21st century, and the first data were published in 2004 [21,36,37]. Primary data were obtained from aquatic ecosystems, and breast milk later began to be used as a biomarker for monitoring human exposure.

Polychlorinated biphenyls (PCBs) are an organic chlorine compound with the formula C_12_H_10−x_Cl_x_ that are classified as persistent organic pollutants. PCBs act as endocrine disruptors and probable human carcinogens. Dichlorodiphenyltrichloroethane, commonly known as DDT, is an organochloride compound originally developed as an insecticide with s strong environmental impact. Commercial DDT represents a mixture of isomers that are metabolized in environment to dichlorodiphenyldichloroethylene (DDE) and dichlorodiphenyldichloroethane (DDD) [38].

The present study is newly focused on detailed monitoring of brominated flame retardants with emphasis on PBDEs and other high-risk substances, such as polychlorinated biphenyls (PCBs) and chlorinated diphenyl ethanes, summarily designated as DDTs, in the fat tissue of the obese middle-aged patients (mostly women) in Czechia who underwent plastic surgery. The sampled fat tissue was analyzed using the sensitive gas chromatography and mass spectrometry technique (GC-MS). The study investigated the current contamination of Czech obese women by halogenated pollutants. The obtained results were compared to the PBDE, PCB, and DDT levels, which were previously published almost a decade ago [39].

## 2. Materials and Methods 

### 2.1. Characteristics of Donors and Taking Fat Tissue Samples

Samples of white fat tissue were taken during surgery at the Clinic of Plastic Surgery of the Královské Vinohrady Faculty Hospital. The entire clinical study was approved by the Ethics Committee of the Third Faculty of Medicine of Charles University. Samples were taken of subcutaneous fat from the breasts (51 samples), abdomen (55 samples), and back (1 sample). A total of 107 samples of human subcutaneous fat were taken (96 women, 11 men); the group has an average age of 43.2 years, in the range from 19 to 76 years. Forty-six individuals were overweight with a body mass index (BMI) of 25.0–29.9 kg/m^2^, while 22 individuals were obese with BMI values of over 30.0 kg/m^2^. Detailed anthropometric data on the age, weight, height, and BMI of the individuals included in the study are summarized in Table 1. In order to obtain the relevant information on possible means of exposure and potential for contamination, the patients completed a questionnaire containing their personal data (age, weight, height), weight changes during the last few months, and their profession (to encompass the possibility of occupational exposure). Blood was also taken from the patients to determine glycemia and the plasmatic lipid level (cholesterol, high-density and low-density lipoprotein (HDL, LDL), and triacylglycerides). After removal, the fat samples were frozen at a temperature of −70 °C and stored for subsequent GC-MS analysis. The tested group is characterized as middle-aged obese patients, mostly women from Czechia that underwent a plastic surgery. This group is comparable by sex, age, and BMI to the liposuction subjects whose exposition to PBDEs, PCBs, and DDTs was monitored in 2009 [39].

### 2.2. Chemicals

The chemical standards employed had a purity of over 98%. The mixture of PCBs indicators contained the 28, 52, 101, 138, 153, and 180 congeners; the individual PCB standards PCB 118 and PCB 112 were purchased from Dr. Ehrenstorfer GmbH (Germany). The individual organopesticide standards: hexachlorobenzene (HCB); α-, β-, γ- isomers of hexachlorocyclohexane (HCH); dichlorodiphenyldichloroethylene (DDE): o, p’-DDE, p, p’-DDE; dichlorodiphenyldichloroethane (DDD): o, p’-DDD, p, p’-DDD; dichlorodiphenyltrichloroethane (DDT): o, p’-DDT, and p, p’-DDT were purchased from Sigma-Aldrich (USA). The individual standards of PBDE congeners 2,4,4’-TriBDE (BDE 28); 3,4,4’-TriBDE (BDE 37); 2,2’,4,4’-TetraBDE (BDE 47); 2,2’4,5’-TetraBDE (BDE 49); 2,3’,4,4’-TetraBDE (BDE 66); 3,3’,4,4’-TetraBDE (BDE 77); 2,2’,3,4,4’-PentaBDE (BDE 85); 2,2’,4,4’,5-PentaBDE (BDE 99); 2,2’,4,4’,6-PentaBDE (BDE 100); 2,2’,4,4’,5,5’-HexaBDE (BDE 153); 2,2,4’,4,5,6’-HexaBDE (BDE 154); 2,2’,3,4,4’,5’,6-HeptaBDE (BDE 183); 2,2’,3,3’,4,4’,5,6’-OctaBDE (BDE 196); 2,2’,3,3’,4,4’,6,6’-OctaBDE (BDE 197); 2,2’,3,4,4’,5,5’,6-OctaBDE (BDE 203); 2,2’,3,3’,4,4’,5,5’,6-NonaBDE (BDE 206); 2,2’,3,3’,4,4’,5,6,6’-NonaBDE (BDE 207); and DecaBDE (BDE 209) were purchased from Wellington Laboratories (Canada). Labelled Decabromo diphenyl ether (13C BDE 209) was supplied by Cambridge Isotope Laboratories (USA). The working solutions of standards were prepared in isooctane and stored at a temperature of 5 °C. Ethyl acetate was supplied by Sigma Aldrich (USA). Hexane, isooctane, cyclohexane, and dichloromethane were purchased from Merck (Germany). All the employed solvents were of quality for trace organic analysis. Sulphuric acid (98%) was purchased from Merck (Germany). The stationary phase for gel permeation chromatography, Bio-Beads S-X3 (styrene-divinylbenzene gel, 200–400 mesh) was produced by Bio Rad (USA). Anhydrous sodium sulphate was purchased from Penta (Chrudim, Czechia).

### 2.3. Treatment of Clinical Samples

Human fat tissue (5.0 g) was mixed with 20.0 g of anhydrous sodium sulphate and extracted in a Soxhlet apparatus for 8 h by 170 mL of a hexane/dichloromethane mixture (1:1, ν/ν). The extract was evaporated to dryness at 40 °C, and the dry matter (isolated fat) was weighed to determine the fat content in the tissue. A part of the isolated fat (ca 750 mg) was dissolved in 10 mL of a mixture of cyclohexane/ethyl acetate (1:1, v/v) with addition of the internal standard PCB 112 and purified on a column of Bio-Beads S-X3 with the cyclohexane/ethyl acetate (1:1, v/v) mobile phase. The fraction with a volume of 18–36 mL was collected, evaporated to dryness, and then dissolved in 200 μL of isooctane with addition of the internal standards BDE 37 (5 ng/mL), BDE 77 (5 ng/mL), and 13C-BDE 209 (50 ng/mL), and finally used for GC-MS analysis.

### 2.4. Gas Chromatography with Mass Spectrometry

All the GC-MS analyses were performed on an Agilent 7890A GC (Agilent Technologies, USA) gas chromatograph, connected with a 7000B triple quadrupole mass spectrometer (Agilent Technologies, USA), working in the electron ionizing mode for determining PCBs. Negative chemical ionization was employed to determine BDE. MassHunter quantitative software (version B05.02, Agilent Technologies, USA) was used for processing the data. 

PCBs, organochlorine pollutants and pesticides were determined on a Rxi-17Sil GC column (30 m × 0.25 mm inner diameter, 0.25 μm, Restek, USA); injected volume 8 μL; injection mode: splitless; injection temperature program in a programmable temperature vaporizing (PTV): 50 °C (0.17 min), ramp 600 °C/min to 325 °C (5 min), ramp 100 °C/min to 280 °C; splitless time, flow rate, and pressure: 1 min, 50 mL/min, mobile phase flow rate (He): 1.3 mL/min; oven temperature program: 50 °C (2.67 min), ramp 30 °C/min to 240 °C, 2 °C/min to 270 °C, 40 °C/min to 340 °C (12 min); transfer line temperature: 280 °C. The triple quadrupole worked in the multiple reaction monitoring mode.

PBDEs were determined on a DB-XLB GC column (15 m × 0.18 mm inner diameter, 0.07 μm, Agilent Technologies, USA); injected volume: 2 μL; injection mode: pulse splitless injection; splitless injection time: 1.5 min; injection temperature program in PTV: 80 °C (0.2 min), ramp 600 °C/min to 320 °C (3.5 min); pulseless injection temperature: 50 psi, 1.5 mL/min, mobile phase flow rate (He): 1.3 mL/min, oven temperature program: 80 °C (1.5 min), ramp 30 °C/min to 320 °C (7 min); transfer line temperature: 300 °C. The triple quadrupole worked in the selected ion monitoring mode.

Quality assurance/quality control: Blank human adipose tissue was artificially contaminated (spiked) with all target compounds. Spiking levels were chosen with regard to real-life contamination levels of 2 ng/g for PBDEs, and 10 ng/g for PCBs and DDTs. PBDE, PCB, and DDT recoveries ranged between 76% and 119%, and relative standard deviations were in a range of 5–20%. 

The obtained data were processed statistically using the STATISTICA program. Altogether, data were processed for 107 fat tissue samples. When the measured pollutant concentration was below the limit of quantification of the method (LOQ), half-LOQ was included in the statistical calculation. The determined LOQs for PBDEs are 0.1 ng/g of fat tissue, for PCBs and DDTs, 0.06 ng/g.

## 3. Results and Discussions

### 3.1. Polybrominated Diphenyl Ethers 

Analyses of the occurrence of PBDEs in human fat tissue were originally based on fat tissue obtained by liposuction; now, for the first time in the Czechia, we employed solid fat tissue obtained from healthy volunteers who underwent plastic surgery for aesthetic reasons. In a comparable study performed in the Czechia in 2009, 98 samples of fat tissue obtained from liposuction were analyzed using GC-MS [39]. 

Sixteen common PBDE congeners were monitored in fat tissue. This set of congeners represents the most abundant PBDEs in plastic products, which are usually monitored in the environment, food, and clinical samples [39]. The detailed contents of the individual PBDEs congeners are summarized in Table 2: The largest concentrations were found for BDE 153 and 47, which were detected in 100% of the samples. The next most commonly detected congeners were BDE 99 and 100 in 50%, BDE 183 and 49 in 38% (the most abundant PBDEs are depicted in Figure 2), followed by BDE 154 in 16%, BDE 28 in 12%, BDE 85 in 7.5%, and BDE 66 in 4%. The levels of the other BDE 197, 196, 203, 207, 20,6 and 209 were below LOQ in all the samples. The overall concentration of PBDEs (determined as the sum of PBDEs 153, 47, 99, 100, 183, 49, 154, 28, 85, and 66) varied in the range from 0.05 (1/2 LOQ) to 34.3 ng/g of fat, mean level 3.31 ng/g and median 1.87 ng/g (Table 3). The distribution of the overall levels of PBDEs in the tested group of samples is asymmetric (Figure 3): The greatest number of samples is at the lowest level (0.05 to 1.0 ng/g) and then the number of samples decreases with increasing concentration down to zero for the concentration interval 8.0 to 9.99 ng/g. A total of 6 samples were measured with PBDE concentrations that exceeded 10 ng/g. It follows that a small part of the obese middle-aged women have a greater bioaccumulation potential compared to the rest of the subjects. The determined type of distribution corresponds to a log-normal distribution which can best be described by the statistical parameter of the geometric mean; these values further are summarized in Table 4.

Comparison of these data (Table 2) with the results of the study carried out in Czechia ten years ago indicates that the order of the individual congeners remained the same, but that the overall content of PBDEs decreased from an average value of 4.4 ng/g before 2009 to the current value of 3.31 ng/g. 

### 3.2. PCBs and DDTs

PCBs were manufactured in Czechoslovakia in 1959 to 1984; a measurable concentration was detected among others in milk and the meat of cows in Liblice near Mělník in 1987, and their occurrence in food has been regularly monitored since then (an instruction of the Ministry of Agriculture and Nutrition, ref. No. 2957/87-110). After the production of PCBs was prohibited in 1984, the limits for their contents in foods were reduced. The problem of PCBs is still relevant in Czechia and Slovakia. PCBs are constantly being released into the environment from formerly applied coatings and during burning waste, storage of paints and transformer oil, etc. Consequently, the Czech authorities carry out regular monitoring of PCB concentrations in the atmosphere and foods. The total concentration of PCBs in human fat tissue found in this study varied between 67.5 and 3466 ng/g of fat, with an average value of 776.0 ng/g of fat and median of 563.0 ng/g of fat. Compared to 2009, a decrease is again apparent in the content of most PCB congeners with the exception of PCB 138 (121.6 ng/g fat in 2009, now 193.6 ng/g fat) and PCB 180 (245 ng/g fat in 2009, now 329.1 ng/g fat). PCB 138 and 180 were used most extensively, contaminated the environment, and are constantly being released from old deposits. Consequently, the overall average level of PCBs in fat tissue has increased from 625.5 ng/g before 2009 to the present level of 776.0 ng/g, and the increase is statistically significant at *p*-value < 0.025 (Table 3). The distribution of the overall level of PCBs has a different character compared to that of PBDEs and DDTs. The distribution of PCBs exhibits a maximum at the non-negligible value of 400–599 ng/g, which is almost in the center of the tested interval. From this level, the occurrence of PCBs decreases on both sides of the tested interval. Another alarming fact is that 20% of obese middle-aged women are exposed to very high PCB levels with absolute values greater than 1200 ng/g; see the second maximum on the distribution curve (Figure 4). Thus, the issue of PCBs remains important and requires heightened attention. Moreover, the occurrence of PCBs is several times greater than that of PBDEs. In addition to PCBs, the contents of related substances were determined, such as hexachlorobenzene (HCB) and isomers of hexachlorocyclohexane (HCH), including Lindane. Of these substances, attention should be drawn to the current high content of HCB at a level of 151.5 ng/g (Table 2), which is somewhat higher compared to the value of 120.4 mg/g in 2009.

The concentration of DDTs in fat tissue varies between 11.9 to 2489.3 ng/g of fat, with an average value of 467.4 ng/g of fat and median of 301.5 ng/g of fat. These values are still high, but there has been a demonstrable decrease from 615.6 ng/g before 2009 to the present value of 467.4 ng/g. This 24% decrease is statistically significant at *p*-value < 0.01 (Table 3). The DDT contamination in the environment has a decreasing trend, and it can be assumed that this trend will continue. The distribution of DDTs is also decreasing from the most frequent members at the lowest value of 0–199 ng/g gradually to higher levels (Figure 5) and exhibits a similar character to the distribution of PBDEs.

## 4. Conclusions

This study provides a detailed report of the current occurrence of individual PBDE, PCB, and DDT isomers in human fat tissue obtained from volunteers in Czechia. A total of 107 samples of solid fat tissue taken during plastic surgery performed for aesthetic reasons were analyzed. The obtained data are comparable with the data obtained in a similar study published in 2009, where liquid fat sampled during liposuction was analyzed; however, the analytical part is the same in both studies. The fat tissue was found to contain the following PBDE congeners, in order of decreasing contents: 153, 47, 99, 100, 183, 49, 154, 28, 85, and 66, where the average concentration of the highest content of BDE 153 was 1.02 ng/g of fat. In the majority of the cases, the concentrations of the others were below 0.1 ng/g of fat. The overall concentration measured as the sum of 10 congeners attained an average value of 3.31 ng/g and varied from 0.05 to 34.3 ng/g, which is, on an average, 25% less than was found in human fat tissue 10 years ago. The overall level of PBDE contents corresponds to only 0.4% of the overall concentration levels of PCBs and 0.7% of the overall levels of DDTs. The contamination of obese middle-aged women in Czechia by PBDEs is thus lower than their contamination by PCBs and DDTs. 

Unfortunately, more extensive comparison of data for Europe or the world is greatly limited. The closest available data that can be used for at least an approximation consist in the concentrations of PBDEs determined in blood or breast milk, for which certain trends can be followed. If we strictly concentrate only on the occurrence of PBDEs in fat tissue [28,30,40,41,42,43,44,45,46], then we find that various results were obtained in various studies, and the conclusion that follows from them is that BDE 153 and 47 are the main congeners in human fat tissue. However, all these studies were based on analyses of fats obtained from cadavers during autopsies. It follows from comparison of the data obtained here with studies analyzing the occurrence of PBDEs in breast milk [47] that the concentrations of the individual congeners are similar, but the order of their contents differs, which could be connected with metabolic changes during breast feeding that affect PBDE metabolism. Similar changes in the order of occurrence of the metabolites were found in other clinical and environmental studies [48,49,50].

The concentration of PCBs in human fat tissue increased compared to 2009 for only two congeners, 138 and 180, where the concentrations of the other ones are lower. However, summarily, the average PCB level increased from a value of 625.5 ng/g ten years ago to the current level of 776.0 ng/g, an increase of 24%. In addition, the PCB concentration levels are more than a hundred times higher than those of PBDEs. On the other hand, the average contents of DDTs decreased from 615.6 ng/g ten years ago to the present value of 467.4 ng/g. 

## Figures and Tables

**Figure 1 ijerph-16-04105-f001:**
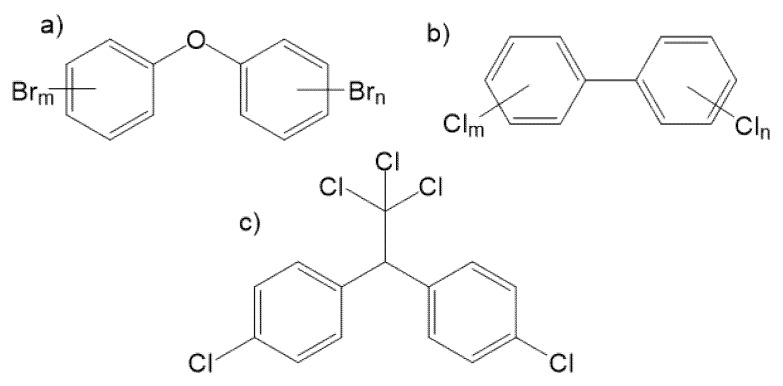
General structure of (**a**) polybrominated diphenyl ethers (PBDEs), (**b**) polychlorinated biphenyls (PCBs), and (**c**) dichlorodiphenyltrichloroethane (DDT).

**Figure 2 ijerph-16-04105-f002:**
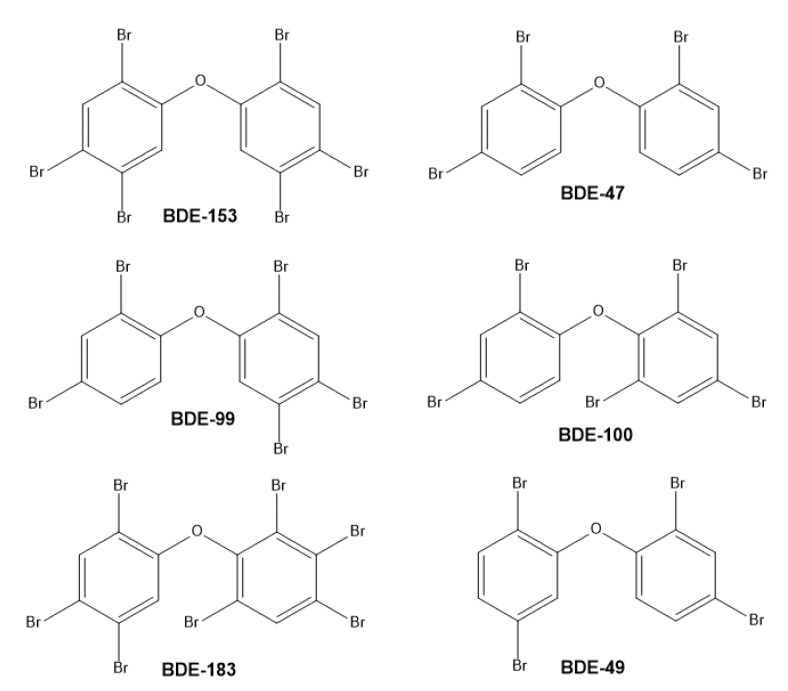
The commonest PBDE congeners found in human fat tissue.

**Figure 3 ijerph-16-04105-f003:**
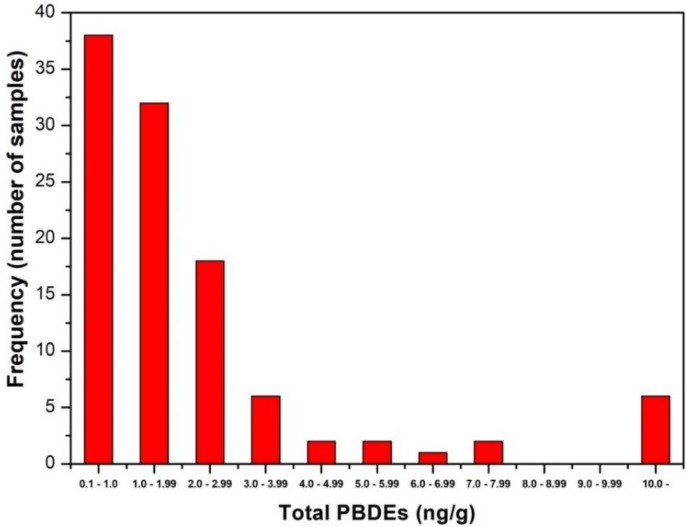
Frequency distributions of the concentrations of PBDEs in human fat tissue.

**Figure 4 ijerph-16-04105-f004:**
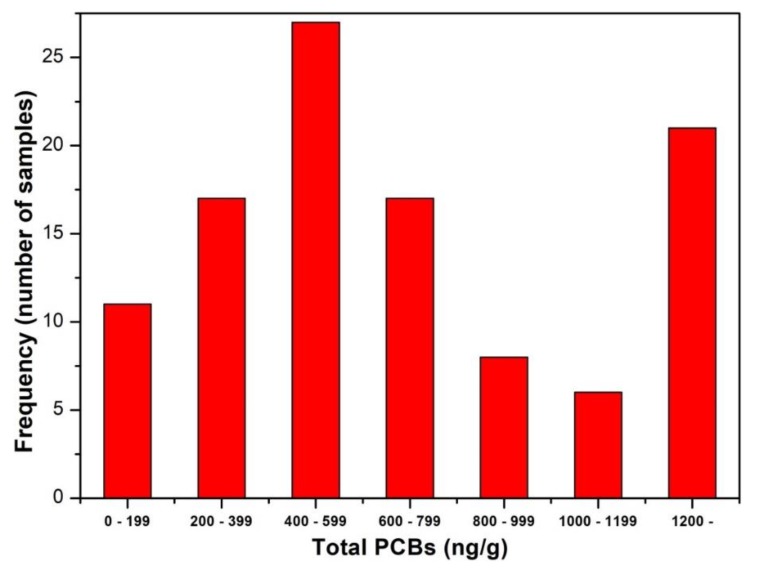
Frequency distribution of the concentrations of total PCBs in human fat tissue.

**Figure 5 ijerph-16-04105-f005:**
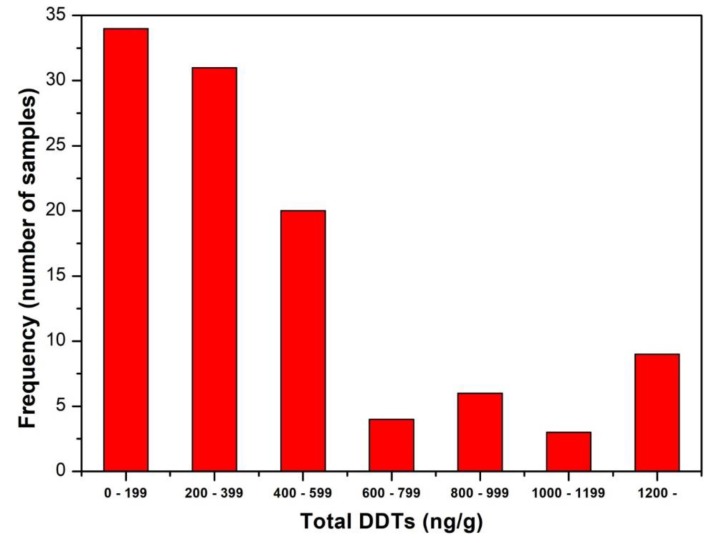
Frequency distribution of the concentrations of total DDTs in human fat tissue.

**Table 1 ijerph-16-04105-t001:** Anthropometric parameters of the group of 107 volunteer fat tissue donators. Standard deviation (SD), interquartile range (IQR), minimum (Min), and maximum (Max) value.

Variable	Mean	SD	Median	IQR	Min	Max
Age (years)	43.2	12.4	41.0	16.5	19.0	76.0
Weight (kg)	75.2	15.2	74.0	18.0	49.0	150.0
Length (cm)	167.4	7.27	166.0	6.0	150.0	190.0
BMI (kg/m^2^)	26.8	4.88	26.0	6.0	18.7	47.3

**Table 2 ijerph-16-04105-t002:** Contents of brominated flame retardants and organochlorine pollutants in human adipose tissue (ng/g lipid in adipose tissue).

Analyte	Mean	CV	Median	IQR	P5	P95	Min	Max
BDE 47	0.59	1.64	0.30	0.56	0.05	2.47	0.05	7.03
BDE 49	0.22	3.04	0.03	0.16	0.05	0.80	0.05	6.33
BDE 99	0.49	2.46	0.14	0.24	0.05	2.18	0.05	8.23
BDE 100	0.19	1.37	0.11	0.17	0.05	0.76	0.05	1.66
BDE 153	1.02	1.78	0.67	0.64	0.10	2.05	0.10	16.6
BDE 183	0.70	1.93	0.10	0.29	0.10	3.73	0.10	7.91
PCB 28	1.88	2.32	0.99	1.52	0.03	4.03	0.03	35.5
PCB 52	0.16	1.22	0.07	0.19	0.03	0.47	0.03	1.06
PCB 101	0.61	1.31	0.33	0.66	0.03	1.93	0.03	4.34
PCB 118	12.9	0.85	9.51	10.9	2.59	31.9	1.60	68.8
PCB 138	193.9	1.05	142.6	155.7	33.2	530.3	11.4	1473
PCB 153	237.6	0.89	177.5	254.3	35.7	577.8	19.0	1114
PCB 180	329.1	0.84	241.4	244.7	53.2	827.3	20.9	1652
HCB	151.5	1.32	65.6	127.9	11.4	533.2	8.68	1152
α-HCH	0.53	1.06	0.49	0.68	0.03	1.21	0.03	3.87
β-HCH	29.5	1.26	14.64	30.8	2.70	127.0	1.40	190.4
Lindane	0.39	0.74	0.44	0.66	0.03	0.71	0.03	0.89
o,p’-DDE	0.37	1.02	0.24	0.70	0.03	0.74	0.03	2.48
p,p’-DDE	409.6	0.99	252.3	363.7	70.9	1267	45.5	2217
o,p’-DDD	0.47	1.07	0.23	0.85	0.05	0.94	0.05	3.39
p,p’-DDD	10.0	2.35	2.73	7.41	0.25	32.63	0.03	165.6
o,p’-DDT	1.41	1.61	0.87	1.25	0.20	4.25	0.11	18.6
p,p’-DDT	45.5	1.80	25.6	36.7	4.15	133.9	0.50	617

CV—coefficient of variation, IQR—interquartile range, P5—percentile 5%, P95—percentile 95%, Min—minimum; and Max—maximum value.

**Table 3 ijerph-16-04105-t003:** Overall level of PBDEs, PCBs, and DDTs in human fat tissue in ng/g of fat and comparison with previous studies for the Czech Republic published in 2009 [39].

	Current Study (*n* = 107)					Previous Study (*n* = 98)					*p*-value
Analyte	Mean	CV	Median	P5	P95	Mean	CV	Median	P5	P95	
PBDEs^1^	3.31	1.26	1.87	0.45	12.2	4.4	1.39	3.1	0.9	9.1	>0.05 <0.1
PCBs^2^	776.0	0.80	563.0	161.9	2136	625.5	0.6	595.0	214.6	1285	<0.025
DDTs^3^	467.4	0.94	301.5	108.6	1502	615.6	0.71	509.7	137.7	1668	<0.01

^1^ Sum of PBDE congeners 47, 153, 99, 100, 183, 49, 154, 28, 85, and 66, ^2^ sum of congeners 28, 52, 101, 118, 138, 153, and 180, ^3^ sum of dichlorodiphenyldichloroethylene (DDE), dichlorodiphenyldichloroethane (DDD), and DDT isomers, *n*-group size, CV—coefficient of variation, P5—percentile 5%, P95—percentile 95%, *p*-value—probability value.

**Table 4 ijerph-16-04105-t004:** Contents of the main PBDEs and organochlorine pollutants in human adipose tissue (ng/g lipid), Geometric means and 95% confidence intervals with lower (LCI) and upper (UCI) border.

Analyte	Geometric Mean	LCI	UCI
BDE 47	0.27	0.22	0.35
BDE 99	0.15	0.12	0.20
BDE 100	0.11	0.10	0.14
BDE 153	0.65	0.55	0.77
PCB 28	0.72	0.54	0.58
PCB 52	0.079	0.063	0.100
PCB 101	0.24	0.18	0.32
PCB 118	9.33	8.01	10.9
PCB 138	134.2	113.6	158.5
PCB 153	162.8	136.4	194.2
PCB 180	247.4	213.1	287.2
HCB	73.7	58.4	92.9
α-HCH	0.26	0.19	0.34
β-HCH	15.6	12.5	19.4
Lindane	0.20	0.15	0.27
o,p’-DDE	0.17	0.13	0.22
p,p’-DDE	282.0	239.0	331.9
o,p’-DDD	0.21	0.16	0.28
p,p’-DDD	3.06	2.24	4.19
o,p’-DDT	0.78	0.64	0.95
p,p’-DDT	24.4	19.5	30.4

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
