# Peer review of "Evaluation of the Burdening on the Czech Population by Brominated Flame Retardants"

_ijerph, 2019, doi:10.3390/ijerph16214105_

Round 1

Reviewer 1 Report

Comments to the manuscript ijerph-534532. Title: "Evaluation of the burdening on the Czech population by brominated flame retardants" for the International Journal of Environmental Research and Public Health.

This manuscript is about a study on several families of persistent organic pollutants (POPs) in adipose tissue of people from the Czech Republic. Among them, the most representativeis the congeners of polybrominated diphenyl ethers (PBDEs) and polychlorinated biphenyls (PCBs), and isomers of pesticide DDT (DDTs). The authors have analysed a part of the most common or indicator congeners or isomers by GC-MS and results are shown and treated by families. Besides, the authors have compared them to those obtained in a former study in the same population in 2009. Anyway, I have some comments to do.

Typewriting mistakes or doubts about them.

In my opinion, when talking about the family of PBDE, the authors should use PBDEs. The same for PCBs, because it is a generic name to talk about 209 different congeners in both. In the case of DDT and its isomers, I also think it would be better to talk of DDTs. They are the common names for all of them. I have found a mistake in three different parts of the text. Please, change BGE into BDE: lines 198, 174, 182. Line 237. Please, change the name of lindane. In my opinion, it is not necessary to write its name in italics.

Abstract.

Please, remove the headings like background, methods, etc. and try to resume the abstract to follow the rules for authors including no more than 200 words. At this moment, I think there are 268 words. Besides, please rewrite it to have an appropriated form.

Introduction.

I understand the use of abbreviation BDE, but these compounds are usually called PBDEs or even BDEs. PBDEs are introduced with information about chemical structure, bromine-sustitution, isomers, and more important individual congeners. However, I miss something similar about the other family of compounds, PCBs and DDTs. Line 39. About inorganic retardants and their potential risk, I would like you could add a reference. According to the manuscript, phosphorous is one of the considered substantial burdens for the environment and a potential health risk for human. So, this should be taken into account because one class of inorganic retardants belong to the phosphorous category. In line 70, authors talk about other brominated flame retardants where they include the lindane. That is a mistake because its structure only has to do with chlorine.

Materials and methods.

I understand that these samples are very valuable and not easy to find and to overcome Ethical Committees. I suppose that the main limitation to obtain a more balanced rate of female and male adipose tissues is its origin, coming from main volunteers after plastic surgery. About chemicals paragraph: Please, revise the names of HCH, DDE, and add the chemical name for DDT. Authors include some compounds like HBB, PBT, PBEB, BTBPE, octabromotrimethylphenylindane and decabromodipheylethane as BDE standards, but they finally do not talk about them along the text. Also, these last compounds are included as BDE standards when HBB, PBT, PBEB, BTBPE, octabromotrimethylphenylindane do not have a BDE structure. Lines 108 and 111. Please revise and change the following congener formulas: PBDE 37 (3,4,4′-TriBDE) and 183 (2,2',3,4,4',5',6-HeptaBDE).

Results and discussion.

In my opinion, the first paragraph of section 3.1 should move to the introduction section, because they tell us about the health consequences of PBDEs in several human tissues referenced in different articles. As the authors talk about a study about monitoring in the population of Czech Republic published in 2004, I consider that they should be taken as background. In line 184, authors say that it was not possible to discover a direct dependence between BDE high levels and life style. According to the questionnaire, I think that is not possible to find an answer to that question because nothing was asked related to that subject. I think that it would have been interesting to ask about food habits, private address, places where studying or working and if people travel daily. PBDEs are very common in many technical devices and homeware very common at homes and that would have been very interesting to introduce. From what it is said about the questionnaire I am not able to conclude if those questions were formuled. The table 2 shows the contents of individual congeners or isomers analysed with high values. Although nothing is said, I suppose that PCB 118 was included because it was already analysed in the reference previous study. However, PCB was one of the indicator PCBs and was removed from them and better named as oine of the mono-ortho dioxin-like PCBs. If so, please, add a short comment about this circumstance. Table 2. As in table 1, please, add the meaning of Min and Max. Table 3. It could be convenient to clearly group columns belonging to Current study and those to the Previous study. Line 196. From data appearing in table 3, it is not possible to deduce the order of individual congener levels because only totals are shown. Line 239. Related to the comparison of HCB levels in this study and the previous one, the results are 151.5 ng/g for 2019 and 120.4 ng/g for 2009. I understand that may be there is an increase, however, they are different types of means: arithmetic and geometric. The result for HCB in 2009, according to table 4, is a geometric mean of 73.7 ng/g instead of 120.4. In order to know the comparison, I understand what the authors say about cadavers. Anyway, and with all the reservations, they could be used to that goal. However, although there are not many articles, you can find results in these following articles where samples were taken from alive people: She et al. Chemosphere (2002) 46(5): 697-707. Kunisue et al. Environ. Int. (2007) 33: 1048-1056. Artignac et al. Environ. Pollut. (2009) 157(1): 164-173. Petreas et al. Environ. Int. (2011) 37(1): 190-197. He et al. Environ. Res. (2018) 167: 160-168.

Conclusions.

Line 253. DDT has isomers instead of congeners.

References.

Please, add a ‘.’ Between the last author and the title of all the articles. Please, write the journal name in the abbreviated form in reference 24: Environ. Contam. Toxicol.

Author Response

Thank you for your rational comments to our manuscript. All recommendations were considered and the manuscript was rewritten accordingly. All changes are highlighted and written in red. In details:

Typewriting mistakes: Your recommendation was accepted and abbreviations PBDEs, PCBs and DDTs are used in whole text. Also typographic mistakes as “BGE” and “lindane” were corrected. Abstract: Headings were removed and the text was shortened to 200 words. Introduction: The basic information about PCBs and DDTs were newly added to text and their structures were depicted in Figure 1. A new reference about inorganic retardants was added and also the class of phosphorous retardants was mentioned. Line 70: the description “brominated” was removed. Materials and methods: The group of tested volunteers was better described as a group of middle-age obese women. All suggested typographical changes for chemicals were accepted. Results and discussions: The first paragraph of the results section was moved to the introduction section. PCB 118 was was analysed in current study and its concentration is mentioned in Table 2 (12.9 ng/g). Line 184: Discussions about the dependence between PBDEs high levels and life style was removed. Additional description was added to the caption of Table 2 and also Table 3 was improved. Line 196: Corresponding data are depicted in Table 2; correction was made. Line 239: The result for HCB is performed correctly and arithmetic means are compared in both years. All other minor comments were fully accepted and also suggested citations were added to text. Conclusions: The term “isomers” is used instead of “congeners”.

References: Abbreviated journal form in reference 24 is used and a “,” between the last author and the title of all articles was omitted.

Reviewer 2 Report

The investigators are providing helpful public health information regarding background exposure to a number of POPs for the Czech Republic.  The data should be published for this reasons.  However,  before I can recommend acceptance for publication I'm recommending the following High Level changes:

Remove all risk/causation statements – most of the alleged associations that are linked to low background exposure come from epidemiological studies that are questionable and cannot establish causation,(e.g., cross sectional studies that lack dose-response data corroborating cause and effect).  Specifically, Lines 162 to 164 must be removed or edited to clarify the hypothetical nature of these reported association.  The biomonitoring data themselves are important and there is no need to include controversial public health opinions in the manuscript. The study is small, involves a very specific subset of mostly Czech women (obesity-related plastic surgery cases) and does not represent the general Czech population.  Because of this, all of the apparent trends conclusions must be softened and not extrapolated to the general Czech population. Because these subjects are apparently obese middle-aged women (for the most part) the authors must limit their conclusions on representation and trends to just this subset of Czech subjects.  The authors must establish that the demographics of their study (mostly women with an average age of 43, weight and BMI are comparable to the liposuction subjects whose results were published in the early 2009 paper.   This will help to support the comparison interpretations between these two sample populations.   The authors (line 267) point out that fat samples are not common for making their comparisons over time while they do not address the fact that lipid-adjusted serum/blood results are correlated with fat sample results. Can’t the investigators in this manuscript use lipid-adjusted serum results for BDE, PCBs, DDT, etc., that may be available for the Czech republic, for their Conclusion purposes? The manuscript must get the approval of analytical expert peer reviewer since I do not have this expertise. Specific Comments: Needs English Grammar improvements:   Replace all "BGE" with BDE.  This appears to be a typo; line 173 - change the word “content” to “concentration”.  Line 261 “11” – I only counted “10” BDE congeners in the preceding line. Line 260 – more of a curiousity than an information need – is the 0.1 ng/g of fat LOQ stated in line 260 reflective of state of the art analytical sensitivity for these compounds? Line 199: The claim that the apparent downward trend is consistent with the environmental regulations enacted under the Voluntary Emissions Control Action Programme (VECAP) in 2014 is difficult to accept since 2014 and 2018/2019 – the time this study was conducted and the 2014 CECAP, is too small for appreciable declines in POPs with long half-lives in humans. Needs to more explicitly define if all 209 BDEs were analyzed for – this is not clear.  Only 10/209 BDE congeners are stated on line 259 – what happened to the other 210 congeners?   Do the Mass Spec chromatograms resolve all 209 BDE congeners for which only a few exceeded the LOQ? Mention of dust inhalation – this is a minor and insignificant pathway.   For example, oral ingestion of house dust is in the milligram range whereas inhalation only provides for microgram amounts over a 24 hour period.  A better discussion on exposure pathways is needed along with solid citations. Lack of statistics (e.g., table 3 for the 2009 vs. the 2009 results – which represent a very select population of the Czech population).   Are the 2009 and 2019 results statistically different in table 3? Need citations (line 47) to back up the statement that elevated temperatures and intense solar radiation release BDE.   And, how is this reflected in greater house dust, food residuals, and inhalation pathways?

Author Response

Thank you for your rational comments to our manuscript. All recommendations were considered and the manuscript was rewritten accordingly. All changes are highlighted and written in red. In details:

Risk/causation statements were removed from the paper including lines 162 – 164. The studied group was specified as “obese middle-aged women from Czechia” and all conclusions are related to this group. This group is comparable to the liposuction subjects whose results were published in the early 2009 paper as is mentioned in Table 3. Unfortunately we are not able to compare and find correlation between PBDE's concentration in serum/plasma and fat tissue. PBDE's concentrations for plasma/serum are not available in Czechia. We suppose that PBDE's blood concentration should be very low and mostly under the LOQ of our GC-MS method. Available data for Czechia you can see in Scientific Opinion on Polybrominated Diphenyl Ethers (PBDEs) in Food, EFSA Panel on Contaminants in the Food Chain ( CONTAM) European Food Safety Authority (EFSA), Parma, Italy EFSA Journal 2011;9(5):2156. Analytical part of manuscript was corrected, especially names of compounds. All suggested English Grammar improvements were accepted. The sentence related to the Voluntary Emissions Control Action Programme (VECAP) was removed including citation. The determined set of 16 PBDEs was described in more details: Sixteen common PBDEs congeners were monitored in fat tissue. This set of congeners represents the most abundant PBDEs in plastic products, which are usually monitored in the environment, food and clinical samples [38]. The discussion on exposure pathways was added with new citations: The human population is exposed to PBDEs through ingestion of food like fish and shellfish [11,12], meat products and eggs [13,14], dairy products and oils [15]. Instead of food, the ingestion of dust from the home environment represents the highest intake of BDE-209 [16,17]. Infants are exposed to PBDEs via breast milk [18]. Additional exposition pathway represents the inhalation of airborne dust particles in the indoor and outdoor environment [19,20]. Important exposure also occurs during long-term breastfeeding, when PBDEs are released from the fat tissue into the breast milk and blood, similar to PCBs (Figure 1) [21,22]. Statistical data evaluation was performed and summarized in Table 3. New citation related to the releasing of PBDEs from polymeric matrices under radiation was added.

Reviewer 3 Report

This manuscript investigates the change in body burden in white fat of brominated contaminants (BDE, PCB) and other halogenated organic substances (DDT and congeners) by comparing the authors’ present data with data from a similar population published a decade ago.

Critique

This is an interesting paper. The authors were careful and transparent in their method for assessing the comparisons of their data to older published values. This is valuable information.

Specific comments

At line 183, you equate the concentrations of DE in fat tissue to exposure. This is not correct. While the BDE gained entry to the body by exposure, concentrations in fat are due to bioaccumulation. It appears that some individuals may experience greater bioaccumulation potential than others. Thus, there is another aspect to consider. Please re-think this portion of the paper and replace this text.

At lines 264-265 of the conclusions, it is not clear what you are trying to say. Please clarify.

Minor comments

Lines 35-38: This is a long, run-on sentence. Consider changing to read: “…and Japan. They were…risk of fires [1]. Their use…”

Line 40: Please add a comma: “health, and organic retardants…”

Line 44: Please change text to read: “…consists of polybrominated…”

Lines 65-66: This sentence is confusing. Do you mean to say: “These substances have a tendency to accumulate,…”?

Lines 66-67: The information about breast feeding is redundant with the sentence at line 62. The breast feeding information should be deleted from one of these 2 locations.

Line 68 (and throughout): “Plasmatic” should be replaced with “plasma”

Line 73: Please replace “the submitted study” with “The present study”

Line 77: Please replace “was concerned with discovering” with “investigated”

Lines 78-80: Consider changing the text to read: :…halogenate pollutants compared to the previous levels of PCB and DDT, which were previously published almost a decade ago [16].”

Lines 83-85: This is a very long sentence. Consider Changing the text to read: “…Faculty Hospital. The entire…”

Line 86: Do you mean “abdomen” rather than “stomach” (which is an internal organ)?

Line 88: At the start of a sentence, numbers should be spelled out. Please change “46” to “Forty-six”

Line 94: Please change “professional” to “occupational”

Line 101: Consider changing the text to: “The chemical standards employed…”

Line 26: Consider changing the text to read: Human fat tissue (5.0g) was mixed…”

Line 129: Please change “cca” to “ca”

Lines 138-139: Consider changing the text to read: “…determining PCB. For organochlorine pollutants, negative chemical ionization was employed to determine BDE.”

Lines 155-158: This is a very long sentence. consider changing text to: “…fat tissue samples. When the measured…”

Line 161: Please delete “demonstrably”

Line 166: Consider changing to “…ecosystems; breast milk…”

Line 167: Consider changing text to: Analyses of BDE in human tissue…”

Line 173: Please replace “found” with “detected”

Line 173-174: Consider making 2 sentences: “…100% 0f samples. The next most commonly detected contaminants were BDE 99…”

Line 174: You have written BGE here and in multiple places in the reminder of the paper. Do you mean BDE? Please check the document carefully and correct this.

Line 175: Please replace “follow” with “followed by”

Line 175: Please separate into 2 sentences: “…in 4%. The levels…”

Line 177: Consider enclosing within parentheses the text “…determined as…85 and 66”

Line 180: Consider changing the text to: “…lowest level (0.05 to 1.0 ng/g) and…”

Line 181: Please end the sentence at the end of this line.

Line 182: Please replace the text :this is followed…6 samples.” with “A total of 6 samples was measured with BDE concentrations that exceeded 10 ng/g.”

Line 217: Why is “and the hygiene rules were updated in 1996” int the text? What were the changes? Did they have an impact on this study? If this is not important, delete the text. If it is important, please expand the discussion of the changes.

Line 218: Please make into 2 sentences: “…Czech Republic. PCB are constantly…”

Line 245: Please change text to read: “The DDT contamination in the environment…”

Line 254: Please replace “undergone” with “performed”

Author Response

Thank you for the very positive evaluation of our manuscript. All your comments were considered and the manuscript was rewritten accordingly. In details:

The bioaccumulation potential of individuals was mentioned as the cause of the high PBDEs level: A total of 6 samples were measured with PBDEs concentrations that exceeded 10 ng/g. It follows that a small part of the obese middle-aged women have a greater bioaccumulation potential compared to the rest of subjects. The sentence in ln. 264 – 265 was rewritten as: The contamination of the obese middle-aged women in Czechia by PBDEs is thus lower than their contamination by PCBs and DDTs. All minor comments were fully accepted as you can see in the revised paper.

Reviewer 4 Report

Current occurrence of the individual BDE, PCB and DDT congeners in human fat tissue obtained from volunteers in the Czech Republic was investigated in this study.
Comments are as follows:
1. The English grammar of the paper should be carefully revised. e.g., “of PCBs from 625.5 ng/g, published in 2009, to the current level of 776 ng/g. On the other hand, the level of DDTs decreased and currently has a value of 467.4 ng/g, which is 24% less than in 2009. Conclusions: The contamination of the population by more modern types of pollutants, such as BDEs, is incomparably lower than that by PCB and DDT and is also decreasing” (line 27-30);
What do “they” mean? (line35)ï¼›Similarly, the expression is not clear,such as “their use” (line37),“which they” (line42); “by gas chromatography with mass spectrophotometric detection” (line 21) should be“by gas chromatography and mass spectrophotometry”ï¼›“see the second maximum on the distribution curve (Figure 4)” (line 234).
2.The title of the paper needs to be revised. Can 107 samples represent Czech Republic?
3.“. Results and discussions” are ambiguous and need to be revised.
4.Why didn't BDE183, 49,154,28,85,66 appear in Table 2?
5.“Consequently, the overall average level of PCB in fat tissue has increased from 625.5 ng/g before 2009 to the present level of 776 ng/g; a slight decrease can be observed when the medians are compared: 625.5 ng/g before 2009 and 563 ng/g at the present time (Table 3) (line 226-229).” This passage is contradictory.
6.The conclusion is too long.

Author Response

Thank you for your rational comments to our manuscript. All recommendations were considered and the manuscript was rewritten accordingly. All changes are highlighted and written in red. In details:

The English grammar of paper was improved and all suggested points were accepted. The paper title was revised as: Burdening of the obese middle-aged women by brominated flame retardants in Czechia. Also the results and discussions section was completely revised. All most abundant PBDEs are newly added to Table 2. Suggested lines 226 – 229: “a slight decrease can be observed when the medians are compared: 625.5 ng/g before 2009 and 563 ng/g at the present time” were removed from the text.

Round 2

Reviewer 1 Report

Comments to the manuscript ijerph-619589-v2. Former title: "Evaluation of the burdening on the Czech population by brominated flame retardants". New title: “Burdening of the obese middle-aged women by brominated flame retardants in Czechia” for the International Journal of Environmental Research and Public Health.

In this second version of the manuscript, I have noticed that authors have followed most of the comments I gave in my frist revision. However, I consider that some more must be done.

In all the text, please, remove the ‘s’ in ‘PBDEs’, ‘PCBs’, and ‘DDTs’ in the following lines: 18, 19, 21, 22, 116, 121, 177, 190, 199, 209, 222, 225, 234, 246, 255, 259, 264, 277, and 283. In my opinion, and to talk in a uniform way, please change the name into PBDEs in line 25, 208 (table 3), 237, and 284. Line 128. Please, change the name of the labelled compound as follows: ‘Labelled Decabromo diphenyl ether (13C BDE 209)’. Line 153. When talking about PCBs and organochlorine pollutants, this last should be ‘organochlorine pesticides’ because PCBs are also organochlorine. Lines 158 and 164. Please separate into two words: ‘transfer line’. Line 171. Please, separate the name: Polybrominated diphenyl ethers. Line 186. I consider ‘range’ a better statistical word than ‘interval’. I am not sure if the name of Czechia is right in lines 33 and 219. Please, revise if it needs the article in these cases.

Title.

According to the samples taken in this study, I cannot accept the new title, because how can be explained that 11 samples came from men? (line 99). The same for the abstract in line 15. This study is not centered in obese middle-aged women, because there are samples from 11 men and also 46 people were overweighted and 26 were classified as obese. But what happened to the other 39 people? (lines 100-102). They cannot be classified in any of both classes. I consider that the former title better reflected the study. Line 238. Complete the name of HCH: hexachlorocyclohexane.

Introduction.

The introduction that authors have added at the end of this section is right, but, in my opinion, it should be moved to the line 79, just before the text about the focus of the present study. Please, pay attention to the change in the order of the references after moving. Line 70. Lindane is regulated in the Stockholm Convention, but cannot be considered as a PBDE or a flame retardant.

Materials and methods.

Line 117. What has been used the 2,2’,4,4’,5,5’-hexabromobiphenyl (PBB 153) for? I am not able to find it in the text except in this line. About chemicals paragraph, please, add the chemical name for DDT (dichlorodiphenyltrichloroethane). Line 150. Please, remove the “For organochlorine pollutants”’ form the text in order to this sentence has sense. GC-MS. Please, revise and change the following parameters: Lines 155 and 162. Although it is not written, I suppose that the injection temperature program was used in a Programmable Temperature Vaporizing (PTV). So, the 600 °C/min is a correct value? Is the 8 µL correct too as the volume of injection and with just 1 min of vent time? In this same part, are you sure of the ramp of temperatures? After the temperature went to 325 °C the new ramp was 100 °C to go to 208 °C. Something is wrong in this part. Lines 154: When talking about injection mode, I suppose that you are talking of splitless. Besides, what you call as ventilation time and pressure I consider they should be written as splitless time and splitless flow. Please, add a short comment about blanks (to discard crosscontamination of samples) and procedure recoveries of the three families of compounds.

Results and discussion.

Line 181. Much better to use ‘congeners’ instead of contaminants to talk about the next coomonly detect contaminants. Line 24 (abstract). It is a very tiny detail, but you say that DDT value in this present study is 467.4 ng/g but in table 3 that value is 467.3. Please, revise this and the same result appearing in lines 244, 245, and 288. In a similar way, and according to table 3, the result in lines 229 and 282 should be 776.0 ng/g. In table 3, please, change the median for PCBs into 563.0 ng/g as it appears in line 224. In line 240, I consider that the ‘(Table 2)’ should be moved to appear just after the value of 151.5 ng/g, because it could look like that the value from 2009 is also shown in that table 2.

References.

Please, complete the authors of reference number 2. There are two authors not included: Van den Ende, N.; Van der Veen, I. Please, complete the name of the journal in reference 33: Environ. Contam. Toxicol.

Author Response

Dear reviewer 1,

Thank you for your rational comments to our manuscript. All recommendations were fully accepted and the manuscript was rewritten accordingly. All changes are highlighted and written in red. In details:

“s” is removed in the suggested lines. All other typographical changes were performed. Title: The former title is used: “Evaluation of the burdening on the Czech population by brominated flame retardants” and abstract is also changed. Name of HCH is corrected. Introduction: Text about PCBs and DDTs is moved and term “pesticides” is added to line 70. Materials and methods: chemical PBB 153 is removed, full name of DDT is added, line 150 is corrected and description of GC-MS is improved. Results and discussion: All suggested recommendations are accepted. References: Two authors are added to reference 2 and the name of the journal is completed in reference 33.

Reviewer 4 Report

The conclusion should not be too long, and the key points should be highlighted.

Author Response

Dear reviewer 4,

Thank you for your comments to our manuscript.

Two sentences are removed from conclusion and last two paragraphs are combined. Also main conclusions are highlighted and written in italics.